# The impact of diabetes on visual acuity in Ethiopia, 2021

**Mulu Tiruneh Asemu** **\*, Mengesha Assefa Ahunie**

Department of Public Health, College of Health Sciences, Debre Tabor University, Debre Tabor, Ethiopia

* tirunehmulu1@gmail.com

**Data Availability Statement:** All relevant data are within the manuscript and its Supporting information files.

**Funding:** No specific funding for this work.

## Abstract

### Background

Diabetes mellitus is a complex metabolic disorder characterized by hyperglycemia that results from defects in insulin secretion, insulin action, or both. Glaucoma is the ocular complication of diabetic illness. In addition to this, retinopathy, maculopathy, ischemic optic neuropathy, extra-ocular muscle palsy, iridocyclitis, and rubeosis iridis were other complications. This study aims to determine the impact of diabetes on visual impairment and blindness among diabetic patients in Ethiopia.

### Methods

This hospital-based cross-sectional study includes 401 samples of diabetic patients in the University of Gondar Comprehensive Specialized Hospital from January 2017 to January 2019. The multinomial logistic regression model was employed to identify significant differences among the factor variables.

### Results

The magnitude of blindness was 32.17%, and the burden of severe visual impairment was 12.46%. Of the total patients, 120(29.9%) were have diabetic retinopathy of whom, 113(94.2%) were blind either in the right, left, or both eyes and 3 (2.5%) had severe visual impairment. One hundred twenty-six (31.42%) patients developed diabetic maculopathy of whom, 117 (92.85%) were blind either in the right or left eye, and one (0.8%) had severe visual impairment. From the whole diabetic patients, the magnitude of glaucoma was 186(46.38%), and from the patients who developed glaucoma was blind visual impairment 127(68.27%) either in the right or left eye. Thirty-eight (20.34%) had severe visual impairment. Glaucoma was significantly associated with severe visual impairment and blindness (p<0.001). Glaucoma, diabetic retinopathy, maculopathy, and type of diabetes are factors for visual impairment.

### Conclusion

We found that visual impairment in the category of severe and blindness are frequent in Ethiopian diabetic patients. Glaucoma, diabetic retinopathy, maculopathy are the main predictive factors that determine the occurrence of blindness.

**Competing interests:** No authors have competing interest.

**Abbreviations:** DM, Diabetes Mellitus; WHO, World Health Organization; VI, Visual Impairment; VR-QOL, Vision-Related Quality of Life.

## Introduction

Visual impairment (VI) refers to a functional limitation of the eye or visual system due to a disorder or disease that results in poor vision in the worst eye. According to World Health Organization (WHO) revised definition, it is defined as presenting distance visual acuity worse than 6/18 in the worst eye [1]. Classification of severity of visual impairment recommended by the Resolution of the International Council of Ophthalmology and WHO Consultation includes Moderate VI, Severe VI, and blindness based on presenting visual acuity worse than 6/18, 6/60, and 3/60 respectively [1, 2]. Among the global population, 216.6 million were moderate or severe visual impairment. The leading causes were uncorrected refractive error (116·3 million), cataract (52·6 million), age-related macular degeneration (8·4 million), glaucoma (4·0 million), and diabetic retinopathy (2·6 million) [3]. Visual impairment remains a public health problem especially in low and middle-income countries, which were estimated to be four times higher than in high-income countries [4].

Diabetes mellitus is a complex metabolic disorder characterized by hyperglycemia that results from defects in insulin secretion, insulin action, or both [5]. In 2017 there were 451 million adult people with DM globally, a prevalence expected to rise dramatically in the coming decades [6]. Diabetes has become epidemic proportions fuelled by aged people as well as the rapid increase of obesity, extending its greatest impact especially in developing countries [7, 8]. According to different pieces of evidence, diabetic retinopathy is one of the common microvascular complications of diabetes and further could be classified, as proliferative and non-proliferative stages [9]. According to Cheung et al. [10], diabetic retinopathy is the leading cause of preventable blindness. Diabetic retinopathy was also ranked as the fifth most common cause of moderate to severe vision impairment from 1990 to 2020 [3]. Different studies identified that retinopathy was 34.6% prevalent in the population with diabetes compared to 8.8% in those without diabetes [11].

Glaucoma also another leading cause of visual impairment and blindness worldwide, and the burden is expected to increase as the year's increase [12]. Glaucoma is the ocular complication of diabetes besides diabetic retinopathy and maculopathy, and other potentially blinding complications such as ischemic optic neuropathy, extra-ocular muscle palsy, iridocyclitis, and rubeosis iridis [13]. Although vision-related quality of life (VR-QoL) is related to visual field loss [13–16], the impact of visual symptoms on the VR-QoL of glaucoma patients has not been fully elucidated. Sight loss is closely related to old age in the next 20 years, the number of persons aged over 85 years will approximately double, which suggests that also the number of the elderly with visual impairment increase [17].

Currently, there is no baseline data on visual impairment and blindness in diabetic patients in the study setting. So, this study aims to identify the impact of diabetes on visual impairment among Ethiopian diabetic patients.

## Materials and methods

### Study setting and design

A hospital-based cross-sectional study design was employed from January 2017 to January 2019 in the Department of Opthalmology in the University of Gondar comprehensive specialized hospital.

### Source of population

The source population of this study was all diabetic patients referred from the diabetic clinic to the ophthalmology department at the University of Gondar Comprehensive Specialized Hospital.

## Study population

The sample population for this study was all diabetic patients referred from the diabetic clinic for ophthalmologic evaluation for the specified period (from January 2017 to January 2019). Patients who were critically ill and consequently unable to give informed consent for Participation and Patients with incomplete data are not including in this study.

## Sample size determination

The required sample size was computing using single population proportion formula based on the assumption of 95% confidence interval, 5% margin of error, and 26% proportion (p) of visual impairment [8]. An added 10% estimated nonresponse rate made a final sample size of 485.

$$\text{n} = \frac{Z^2 P(1-P)}{d^2} \ , \ \frac{1.96^2 0.26(1-0.26)}{0.05^2} = 462 + 10\% = 485.$$

## Sampling procedure

The sample selecting mechanism for this study was a simple random sampling method in which each of the participants had an equal chance of selection to be part of the study.

## Operational definition

According to WHO classification, patients were categorized based on the vision of a better-seeing eye [7].

✓ Mild visual impairment (near to normal): Presenting distance visual acuity equal to or better than 6/18

✓ Moderate visual impairment: Presenting distance visual acuity worse than 6/18 but equal to or better than 6/60

✓ Severe visual impairment: Presenting distance visual acuity worse than 6/60 but equal to or better than 3/60

✓ Blindness: Presenting distance visual acuity worse than 3/60

✓ Visual impairment: Any person who has poor vision or blindness.

✓ Diabetic retinopathy: Could be categorizing as proliferative or non-proliferative.

✓ Diabetic maculopathy: When the macula sustains some form of damage.

## Data collection and quality control procedure

The structured questionnaire was developing after reviewing relevant works of literature to include all the possible variables that address the objective of this study [8, 18–20]. The data collectors (two data collectors and one supervisor) were recruited from the ophthalmology department to collect the data. A one-day training was given to data collectors on the overall data collection procedure by the principal investigators to ensure data quality. The collected data were reviewed and checked for completeness before data entry could perform. Since this study is prospective, the data collectors explained the purpose of the study to the patients, and the patients participated voluntarily without any additional compensation. At this study period, 401 patients met the inclusion criteria, and they agreed to participate in the study. The remaining 84 patients were excluded from the study because of missing the outcome measure.

After collecting socio-demographic data, other clinical data were obtained during ophthalmo-logic evaluation. The examination was done by an ophthalmologist specializing in glaucoma. The anterior segment was assessed using a slit-lamp biomicroscope. Also, intraocular pressure (IOP) was measured by Goldman tonometry. Presenting the visual acuity was measured using projection charts placed at a distance of 6 m from the patient. Patients who could only count their fingers at 3 m and 1 m attributed a visual acuity of 6/18 and 3/60 correspondingly. The patient is on long-term anti-glaucomatous medication with or without glaucomatous disc changes. Most of the time, patients with visual impairment are not coming for treatment in the early stages of the disease. Due to this reason, when patients come to health care, the problem may reach an advanced stage, and it may be difficult to cure that disease.

## Statistical analysis

Data were analyzed using STATA version 16 software. We described continuous variables using mean and standard deviation (SD) or median, and all categorical variables stated in terms of frequency and percentages. The association between categorical variables was com-puted using the Chi-Square test. Factors associated with visual impairment in the study popu-lation assessed using multinomial logistic regression. We computed univariate and multivariate logistic regression analysis while adjusting for confounders to seek factors influ-encing the development of different causes of visual impairment.

## Ethical consideration

This study was approved by the University of Gondar Ethical Review committee of the Institu-tional Review Board. We obtained verbal informed consent from all the participants.

## Results

On the whole, 401 diabetic patients participated. Of these, 154 (38.4%) were women, and 247 (61.6%) were men. Of the total, 225(56.11%) come from a rural area, and 176(43.89%) patients come from an urban area (Table 1). Of the total patients, 293(73.07%) type 2 diabetes, and the patient's year range was 17 to 90 years with a median of 60 years and their mean of 58.9 years (SD = 14.6). The median duration of diabetes was six years (**Table 1**).

As we have seen from Table 2 below, the magnitude of blindness was 32.17% in either right, left, or both eyes, and the burden of severe visual impairment was 12.47%. Similarly, the visual impairment belongs to moderate accounts for 33.42% of the total patients (**Table 2**).

Among 225 rural diabetic patients, 71(31.5%) were blind, and 28 (12.4%) were severely visual impaired. Similarly, from 176 urban patients, 58(32.9%) were blind, and 22 (12.5%) had severe visual acuity problems (**Table 3**).

Table 4 below indicates that the relationship between visual acuity concerning retinopathy and maculopathy. One hundred twenty (29.9%) participants present with proliferative reti-nopathy. From these, 113(94.2%) were blind either in the right, left, or both eyes. Only 3 (2.5%) had severe visual impairment. One hundred twenty-six (31.42%) patients developed diabetic maculopathy of whom, 117 (92.85%) were blind either in the right, left, or both eyes. As we compare, both diabetic types almost have equal contributions for visual impairment (p<0.000) (**Table 4**).

From the whole diabetic patients, the magnitude of glaucoma was 186 (46.38%). Of the patients who developed glaucoma, 127(68.27%) were blind visual impairment either in the right, left, or both eyes and 38(20.34%) had severe visual impairment. Glaucoma was signifi-cantly associated with severe visual impairment and blindness (p <0.000). The difficulties of visual impairment increase with the severity of glaucoma (**Table 5**).

**Table 1. Sociodemographic and clinical characteristics of study participants.**

| Variables | Category | Frequency (%)/mean(SD) | Percent |
|---|---|---|---|
| Age (years) | Overall | 58.9(14.6) | |
| | Female | 56.7(12.9) | |
| | Male | 60.2(15.4) | |
| Sex | Male | 247 | 61.6 |
| | Female | 154 | 38.4 |
| Place of residence | Urban | 176 | 56.11 |
| | Rural | 225 | 43.89 |
| Type of diabetes | Type 1 | 108 | 26.93 |
| | Type 2 | 293 | 73.07 |
| Hypertension | No | 343 | 85.54 |
| | Yes | 58 | 14.46 |
| Duration of diabetes(years) | Overall | 7.9(5.1) | |
| | Type 1 | 6.3(4.2) | |
| | Type 2 | 8.5(5.2) | |
| Family history | No | 375 | 93.5 |
| | Yes | 26 | 6.48 |

We computed multivariable logistic regression analysis while adjusting for confounders by running univariate analysis to seek factors influencing the development of the different causes of visual impairment (S1 Table). Maculopathy, retinopathy, type of diabetes, and the presence of glaucoma were associated factors for visual impairment in the multivariate logistic regression model. (Table 6).

The multinomial logit estimate comparing type 2 to type 1 diabetes for blind relative to moderate, given the other variables in the model are held constant. The multinomial logit for type 2 diabetes relative to type 1 diabetes is 3.29 units higher for preferring blind to moderate, given all other predictor variables in the model are held constant. In other words, type 2 diabetes is more likely than type 1 to prefer blind to moderate ($\beta = 3.29$, CI = 0.33, 6.25, $p<0.03$). If a person were to increase his diabetic maculopathy, by one unit in the multinomial log-odds for preferring blindness to moderate visual impairment would be expected to increase by 3.24 amount while holding all other variables in the model constant ($\beta = 3.24$, CI = 0.66–5.82, $p<0.014$). For each one-unit increase on this variable, the log-odds of the case falling into the blind visual impairment group (relative to moderate visual impairment group) is increased by 3.77 units. In other words, proliferative retinopathy is more likely than non-proliferative retinopathy to prefer blindness when we compare with those moderate visual impairment($\beta = 3.77$, CI = 0.73–6.82, $p<0.015$). The presence of glaucoma has a greater risk of falling into blindness, and at the lower risk of coming into moderate visual impairment when compared to a person who has no glaucoma ($\beta = 4.91$, CI = 2.86, 6.96, $p<0.0001$).

**Table 2. Frequency distribution of visual acuity.**

| Category of visual acuity | Frequency | Percent |
|---|---|---|
| Normal | 88 | 21.95 |
| Moderate | 134 | 33.42 |
| Sever | 50 | 12.47 |
| Blind | 129 | 32.17 |

**Table 3. Comparison between the place of residence against visual impairment.**

| Category of visual acuity | Rural | Urban |
|---|---|---|
| Normal | 50 | 38 |
| Moderate | 76 | 58 |
| Severe | 28 | 22 |
| Blind | 71 | 58 |
| Total | 225 | 176 |

For each one-unit increase on this variable, the log-odds of the case falling into the normal visual impairment group (relative to moderate visual impairment group) is decreased by 2.5 units. It suggests that a person has type 2 diabetes is at lower risk in the normal visual impairment and greater risk of moderate visual impairment when compared with the patient who has type 1 diabetes ($\beta$ = -2.5, CI = -3.18, -1.83, p<0.0001).

Similarly, a person who has proliferative diabetic retinopathy is at greater risk of falling into severe visual impairment groups and at lower risk of coming into the moderate visual impairment group as compared to patients who have non-proliferative diabetic retinopathy ($\beta$ = 3.55, CI = 0.25, 6.84, p<0.035). The log-odds of the case falling into the severe visual impairment group (relative to moderate visual impairment group) are increased by 3.07 units. It indicates that a patient with a glaucoma case is at high risk of severe visual impairment and less risk of moderate visual impairment as compared with a patient who has no glaucoma case ($\beta$ = 3.07, CI = 2.19, 3.95, p<0.0001) **(Table 6).**

## Discussion

In this study, we obtained that magnitude of visual impairment as blind either in the right, left, or both eyes were 32.17%. The burden of severe visual impairment was 12.47% in either left or right eye. The burden of visual impairment in this study was much higher than Ahmadou et al. [8], Xinzhi et al. [21], and lower than in the study by Somdutt et al. [19]. The difference between this finding and other studies may be due to the age group of the subject or the sample selection, skill of the examiner, method of examination, and the study setting. In our findings, we obtained that diabetic retinopathy, diabetic maculopathy, and glaucoma were statistically significant factors for visual impairment as blind. The magnitude of diabetic retinopathy for this study was (29.9%) which lower than the previous study by Funatsu et al. [22] (37%) and Shibru et al. [20] (51.3%). It is higher than studies conducted in Debre Markos (18.9%) by Melkamu Tilahun et al. [18]. However, it is approximately consistent with Roaeid et al. [23] (30.6%). In our study, the magnitude of glaucoma was (46.38%) which much higher than those studied by Ahmadou et al. [8] (15%). This difference may be due to the study population's varies, and as a result, the

**Table 4. Retinopathy and maculopathy concerning visual acuity.**

| Category Visual acuity | Retinopathy | | Maculopathy | |
|---|---|---|---|---|
| | Proliferative (120) | Non-proliferative (281) | Yes(126) | No (275) |
| Normal | 3 (2.5) | 85 (30.3) | 4(3.17) | 84(30.55) |
| Moderate | 1 (0.83) | 133 (47.3) | 4(3.17) | 130(47.27) |
| Severe | 3 (2.5) | 47 (16.7) | 1(0.8) | 49(17.82) |
| Blind | 113 (93.38) | 16 (5.7) | 117(92.85) | 12(4.36) |
| | Chi-square = 302.1648, p-value <0.000 | | Chi-square = 310.2159, p-value <0.000 | |

**Table 5. Glaucoma and visual impairment.**

| Category of Visual impairment | Presence of glaucoma | |
|---|---|---|
| | Yes (186) | No (215) |
| Normal | 4(2.15) | 84(39.07) |
| Moderate | 17(9.14) | 117(54.42) |
| Severe | 38(20.43) | 12(5.58) |
| Blind | 127(68.27) | 2(0.93) |
| | Chi-square = 281.3725, p <0.000 | |

**Table 6. Factors associated with visual impairment in the multivariable analysis of diabetic patients.**

| Outcome | Variables | Category | Coef. (β) | p-value | 95% CI |
|---|---|---|---|---|---|
| Blind | Sex | Female (R) | | | |
| | | Male | -0.13 | 0.84 | -1.42, 1.15 |
| | Age_cat | 17–39 (R) | | | |
| | | 40–64 | 0.44 | 0.802 | -3.03, 3.92 |
| | | > 64 | -0.64 | 0.722 | -4.17, 2.89 |
| | Type of diabetes | Type 1 (R) | | | |
| | | Type 2 | 3.29 | 0.03 | 0.33, 6.25 |
| | Hypertension | No (R) | | | |
| | | Yes | 0.13 | 0.896 | -1.85, 2.12 |
| | Duration of diabetes | 10 years and above (R) | | | |
| | | Less than 10 years | -0.970 | 0.201 | -2.45, 0.51 |
| | Retinopathy | Non-proliferative (R) | | | |
| | | Proliferative | 3.77 | 0.015 | 0.73, 6.82 |
| | Maculopathy | No (R) | | | |
| | | Yes | 3.24 | 0.014 | 0.66, 5.82 |
| | Presence of glaucoma | No (R) | | | |
| | | Yes | 4.91 | 0.000 | 2.86, 6.96 |
| Normal | Sex | Female (R) | | | |
| | | Male | 0.52 | 0.137 | -0.16, 1.20 |
| | Age_cat | 17–39 (R) | | | |
| | | 40–64 | -0.44 | 0.369 | -1.4, 0.52 |
| | | > 64 | -0.72 | 0.198 | -1.81,-0.37 |
| | Type of diabetes | Type 1 (R) | | | |
| | | Type 2 | -2.5 | 0.000 | -3.18, -1.83 |
| | Hypertension | No (R) | | | |
| | | Yes | 0.01 | 0.99 | -1, 1 |
| | Duration of diabetes | 10 years and above (R) | | | |
| | | Less than 10 years | 0.4 | 0.449 | -0.63, 1.43 |
| | Retinopathy | Non-proliferative (R) | | | |
| | | Proliferative | 1.52 | 0.339 | -1.6, 4.64 |
| | Maculopathy | No (R) | | | |
| | | Yes | 0.31 | 0.779 | -1.83, 2.44 |
| | Presence of glaucoma | No | | | |
| | | Yes | -0.94 | 0.165 | -2.26, 0.38 |
| Moderate | Base outcome | | | | |

*(Continued)*

**Table 6.** (Continued)

| Outcome | Variables | Category | Coef. (β) | p-value | 95% CI |
|---|---|---|---|---|---|
| Severe | Sex | Female (R) | | | |
| | | Male | 0.04 | 0.923 | -0.85, 0.94 |
| | Age_cat | 17–39 (R) | | | |
| | | 40–64 | -0.14 | 0.879 | -1.9, 1.63 |
| | | > 64 | 0.03 | 0.972 | -1.83, 1.9 |
| | Type of diabetes | Type 1 (R) | | | |
| | | Type 2 | 0.7 | 0.284 | -0.58, 1.97 |
| | Hypertension | No (R) | | | |
| | | Yes | 0.005 | 0.99 | -1.33, 1.34 |
| | Duration of diabetes | 10 years and above (R) | | | |
| | | Less than 10 years | -0.77 | 0.132 | -1.77, 0.23 |
| | Retinopathy | Non-proliferative (R) | | | |
| | | Proliferative | 3.55 | 0.035 | 0.25, 6.84 |
| | Maculopathy | No (R) | | | |
| | | Yes | -3.15 | 0.075 | -6.63, 0.32 |
| | Presence of glaucoma | No | | | |
| | | Yes | 3.07 | 0.000 | 2.19, 3.95 |

R: reference group, (visual impairments = moderate is the base outcome).

mean age of the current study is an increase from the previous study. Even if this magnitude seemed higher, it has great importance that an early systematic screening of glaucoma among diabetic patients. The burden of maculopathy was more than two times compared to that studied by Ahmadou et al. [8]. The difference may be due to the duration of diabetes that the average years for this study are 7.9, and 5 years for the previous study. The other reason may be, the presence of hypertension, systolic blood pressure, and diastolic blood pressure were associated with diabetic maculopathy. When we computed a measure of association between visual impairment with retinopathy, diabetic maculopathy, and glaucoma, they have a strong association independently. This result is confirmed by Ahmadou et al. [8] and Somdutt et al. [19]. In this study, gender was not statistically significant for visual impairment. This result is in line with the previous study [8] and [18]. For the current study, type 2 diabetes was a statistically significant factor for visual impairment as blind. The present study shows similar findings with Somdutt et al. [19] and Shibru et al. [20].

Similarly, our study identified that diabetic retinopathy and glaucoma were also associated with severe visual impairment. This result confirms that the difficulties of visual impairment increase with the severity of glaucoma. This finding is in line with the study done in Ethiopia [24]. This may be due to the study population with relatively similar age ranges ($\geq$ 17 years) and the use of a similar cut-off point for defining visual impairment.

## Conclusion

We found that visual impairment in the category of severe and blindness are frequent in Ethiopian diabetic patients. Glaucoma, diabetic retinopathy, maculopathy are the main predictive factors that determine the occurrence of blindness. Glaucoma and diabetic retinopathy are factors that determine severe visual impairment. In addition to this, type of diabetes was the factor of visual impairment.

## Supporting information

**S1 Table. The univariate analysis of visual impairment of diabetic patients.**
(DOCX)

**S1 File. Questionnaire.**
(DOCX)

## Acknowledgments

We would like to extend our gratitude to the patients who participated in this study.

## Author Contributions

**Conceptualization:** Mulu Tiruneh Asemu.

**Data curation:** Mulu Tiruneh Asemu.

**Formal analysis:** Mulu Tiruneh Asemu, Mengesha Assefa Ahunie.

**Funding acquisition:** Mulu Tiruneh Asemu.

**Investigation:** Mulu Tiruneh Asemu, Mengesha Assefa Ahunie.

**Methodology:** Mulu Tiruneh Asemu, Mengesha Assefa Ahunie.

**Validation:** Mengesha Assefa Ahunie.

**Visualization:** Mengesha Assefa Ahunie.

**Writing – original draft:** Mulu Tiruneh Asemu, Mengesha Assefa Ahunie.

**Writing – review & editing:** Mulu Tiruneh Asemu, Mengesha Assefa Ahunie.

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
