## [Decision Letter · Decision Letter 0]

22 Apr 2021

PONE-D-21-05208

The impact of diabetes on visual acuity in Ethiopia, 2021

PLOS ONE

Dear Dr. Asemu,

Thank you for submitting your manuscript to PLOS ONE. After careful consideration, we feel that it has merit but does not fully meet PLOS ONE’s publication criteria as it currently stands. Therefore, we invite you to submit a revised version of the manuscript that addresses all the points raised by the two experts during the review process.

We look forward to receiving your revised manuscript.

Kind regards,

Deepak Shukla

Academic Editor

PLOS ONE

Journal Requirements:

2. Please note that PLOS ONE does not copy edit accepted manuscripts (https://journals.plos.org/plosone/s/criteria-for-publication#loc-5). To that effect, please ensure that your submission is free of typos and grammatical errors.

*Please include additional information regarding the survey or questionnaire used in the study and ensure that you have provided sufficient details that others could replicate the analyses. For instance, if you developed a questionnaire as part of this study and it is not under a copyright more restrictive than CC-BY, please include a copy, in both the original language and English, as Supporting Information.

Reviewers' comments:

Reviewer's Responses to Questions

**Comments to the Author**

1. Is the manuscript technically sound, and do the data support the conclusions?

Reviewer #1: Yes

Reviewer #2: Partly

2. Has the statistical analysis been performed appropriately and rigorously? 

Reviewer #1: I Don't Know

Reviewer #2: Yes

3. Have the authors made all data underlying the findings in their manuscript fully available?

Reviewer #1: Yes

Reviewer #2: Yes

4. Is the manuscript presented in an intelligible fashion and written in standard English?

Reviewer #1: No

Reviewer #2: No

5. Review Comments to the Author

Reviewer #1: This is an interesting article providing insights regarding the status of diabetes induced visual impairment in an Ethiopian county. The authors have collected their data from patients referred to them by the Diabetes clinic and have provided us with detailed explanation about how consent was taken from them regarding data sharing. The article has some interesting findings, such as their patient population had a majority with Glaucoma and related ocular pathologies. They also found that many of their patients fall into the Severe and Blind category.

The results are striking and provides an insight with respect to how a single eye or both eyes were affected due to diabetes or other underlying conditions. The article has good results and the authors have strived hard to put forth their discussions in a large section. However the article mainly suffers from usage of English. I highly recommend that this manuscript be referred to a native English speaker to make appropriate changes such that the article is presented in a more readable manner to the international community. Some further suggestions are made below which might help the authors put forth a stronger case in their article.

A comparison between urban and rural populations need to be shown. How many urban/rural people belong to normal or severe category. Are urban people more or less susceptible to blindness? A good amount discussion is warranted and data needs to be put forward.

The authors do not discuss the medications that the patients currently are on. Are they on any medications? If so why is the visual impairment so bad? Discussion needs to be put forward.

The authors have combined data sets of all age groups 17y to 90y. I would recommend splitting this up into different age groups and performing the same analysis as they performed. In the discussions they do mention that their data is a little different from older publications and that it is because they combined all age groups. I think, the current data set should be subdivided and presented according to age groups. This might be high significance.

Reviewer #2: In the manuscript titled “The impact of diabetes on visual acuity in Ethiopia, 2021” the authors have determined the impact of diabetes on visual impairment and blindness among diabetic patients in Ethiopia and have found that glaucoma, diabetic retinopathy, maculopathy are the main predictive factors that determine the occurrence of poor vision and blindness.

After reviewing the manuscript here are my major concerns:

1. There are extensive grammar and spelling mistakes throughout the manuscript. I strongly recommend the English be corrected professionally so that it is easier to comprehend what the authors are attempting to convey to the reader

2. The authors have distinguished between “blind” and “severe vision” Did they mean “severe vision loss or severe vision impairment”? Authors should add either “loss” or “impairment”

3. The questionnaire was based on “relevant works of literature” but no references were cited

4. The authors have not cited a very similar study in Ethiopia from 2019 - Prevalence of Diabetic Retinopathy and Its Associated Factors among Diabetic Patients at Debre Markos Referral Hospital, Northwest Ethiopia, 2019: Hospital-Based Cross-Sectional Study - Melkamu Tilahun et al 2019.

5. The authors need to be more convincing in explaining the novelty of this study in relation to previously published literature.

6. PLOS authors have the option to publish the peer review history of their article (what does this mean?). If published, this will include your full peer review and any attached files.

Reviewer #1: **Yes: **Tejabhiram Yadavalli

Reviewer #2: No

---

## [Author Response · Author response to Decision Letter 0]

21 May 2021

Response to editors 

1. We prepared the manuscript based on PLOS ONE style templates.

2. We tried to correct the grammar and spelling mistakes.

3. The ethics statement appears in the methods section of our manuscript. We included the questionnaire in the original language and English, as supporting information.

4. We included captions for the supporting information files at the end of the manuscript.

Response to Reviewer 1

1. We have done a comparison between urban and rural in the manuscript.

2. We were discussing the medications that the patient's current status in the manuscript.

3. Based on the reviewer's comment we were splitting up into different age groups and performing the same analysis. 

4. Based on the reviewer’s comment we tried to correct the grammar, spelling, and punctuation mistakes throughout the manuscripts. 

Response to reviewer 2

1. We were trying to correct grammar and spelling mistakes throughout the manuscript.

2. We mean that severe visual impairment, and the manuscript corrected by severe visual impairment.

3. Based on the reviewer’s comment we have cited the reference on the manuscript.

4. Based on the comment we have cited the mentioned paper.

5. We have mentioned the purpose of this study in the manuscript.

---

## [Decision Letter · Decision Letter 1]

7 Jul 2021

PONE-D-21-05208R1

The impact of diabetes on visual acuity in Ethiopia, 2021

PLOS ONE

Dear Dr. Asemu,

Thank you for submitting your manuscript to PLOS ONE. After careful consideration, we feel that it has merit but does not fully meet PLOS ONE’s publication criteria as it currently stands. Therefore, we invite you to submit a revised version of the manuscript that addresses the points raised by the reviewer #2 during the review process.

We look forward to receiving your revised manuscript.

Kind regards,

Deepak Shukla

Academic Editor

PLOS ONE

Journal Requirements:

Reviewers' comments:

Reviewer's Responses to Questions

**Comments to the Author**

1. If the authors have adequately addressed your comments raised in a previous round of review and you feel that this manuscript is now acceptable for publication, you may indicate that here to bypass the “Comments to the Author” section, enter your conflict of interest statement in the “Confidential to Editor” section, and submit your "Accept" recommendation.

Reviewer #1: All comments have been addressed

Reviewer #2: All comments have been addressed

2. Is the manuscript technically sound, and do the data support the conclusions?

Reviewer #1: Yes

Reviewer #2: Yes

3. Has the statistical analysis been performed appropriately and rigorously? 

Reviewer #1: I Don't Know

Reviewer #2: Yes

4. Have the authors made all data underlying the findings in their manuscript fully available?

Reviewer #1: Yes

Reviewer #2: Yes

5. Is the manuscript presented in an intelligible fashion and written in standard English?

Reviewer #1: Yes

Reviewer #2: Yes

6. Review Comments to the Author

Reviewer #1: The authors have addressed all the concerns raised by the reviewer. The Language in the newly added sections seems acceptable but not perfect. If this can be modified before the final publication, that would be great. All the new sentences and paragraph are intelligible and do not stop the reader from understanding the statements.

Reviewer #2: The authors have addressed all the comments and recommendations by the reviewers. However there are still a few minor grammatical and sentence construction issues that could be addressed by a native English speaker.

7. PLOS authors have the option to publish the peer review history of their article (what does this mean?). If published, this will include your full peer review and any attached files.

Reviewer #1: **Yes: **Tejabhiram Yadavalli

Reviewer #2: No

---

## [Author Response · Author response to Decision Letter 1]

12 Jul 2021

Response to Editors 

1. We had reviewed our list of reference, and we tried to correct based on the comments. We did not change the reference lists but, we made complete and corrected the author's name’s 

For reference # 1. We add vol.8

For reference # 2. We add the author's name Dandona R

For reference # 3. We add the author's name Bourne R

For reference #t 4. We add the author's name Flaxman SR

 For reference # 6. We add the author's name Shaw JE

 For reference # 8. We add the author's name Jobert Richie N Nansseu

 For reference # 9. We add the author's name Duran S

 For reference # 10. We add the author's name Mitchell P

 For reference # 11. We add the author's name Andersson D

 For reference # 12. We add the author's name Broman A.T

 For reference # 13. We add the author's name Gedde SJ

 For reference # 14. We add the author's name Willson MR

 For reference # 15. We add the author's name Wang Y

 For reference # 16. We add the author's name Ferarro J

 For reference # 18. We add the author's name Teshome Gobena

 For reference # 19. We add the author's name Girsh G. Kamath

 For reference # 20. We add the author's name’s, Fekadu Aga and Abdisa Boka

 For reference # 21. We add the author's name title MD

 For reference # 22. We add the author's name Suto C

 For reference # 23. We add the author's name Kadiki OA

 For reference # 24. We add the author's name Addisu Wondifraw

2. We had not cited papers that have been retracted. We already used the published articles for our reference lists.

Response to Reviewer 1

Based on the reviewer’s comment we have been addressed all comments previously, and they were ensured that all comments had addressed.

 Response to Reviewer 2

Based on the reviewer’s comment we have been addressed all comments previously, and they were ensured that all comments had addressed.

---

## [Editor Report · Decision Letter 2]

2 Aug 2021

The impact of diabetes on visual acuity in Ethiopia, 2021

PONE-D-21-05208R2

Dear Dr. Mulu Tiruneh Asemu,

We’re pleased to inform you that your manuscript has been judged scientifically suitable for publication and will be formally accepted for publication once it meets all outstanding technical requirements.

Kind regards,

Deepak Shukla

Academic Editor

PLOS ONE
---

## [Editor Report · Acceptance letter]

5 Aug 2021

PONE-D-21-05208R2 

The impact of diabetes on visual acuity in Ethiopia, 2021 

Dear Dr. Asemu:

I'm pleased to inform you that your manuscript has been deemed suitable for publication in PLOS ONE. Congratulations! Your manuscript is now with our production department. 

Kind regards, 

on behalf of

Prof. Deepak Shukla 

Academic Editor

PLOS ONE